# Vitamin K2 Modulates Organelle Damage and Tauopathy Induced by Streptozotocin and Menadione in SH-SY5Y Cells

**DOI:** 10.3390/antiox10060983

**Published:** 2021-06-20

**Authors:** Shruti Shandilya, Kavindra Kumar Kesari, Janne Ruokolainen

**Affiliations:** Department of Applied Physics, School of Science, Aalto University, 00076 Espoo, Finland; shruti.shandilya@aalto.fi

**Keywords:** vitamin K2, tau, ER stress, UPR, mitochondrial dysfunction, SH-SY5Y, Streptozotocin, Menadione

## Abstract

Vitamin K2, known for its antioxidative and anti-inflammatory properties, can act as a potent neuroprotective molecule. Despite its action against mitochondrial dysfunction, the mechanism underlying the links between the protective effects of vitamin K2 and endoplasmic reticulum (ER) stress along with basal levels of total tau protein and amyloid-beta 42 (Aβ42) has not been elucidated yet. To understand the neuroprotective effect of vitamin K2 during metabolic complications, SH-SY5Y cells were treated with streptozotocin for 24 h and menadione for 2 h in a dose-dependent manner, followed by post-treatment of vitamin K2 for 5 h. The modulating effects of vitamin K2 on cell viability, lactate dehydrogenase release, reactive oxygen species (ROS), mitochondrial membrane potential, ER stress marker (CHOP), an indicator of unfolded protein response (UPR), inositol requiring enzyme 1 (p-IRE1α), glycogen synthase kinase 3 (GSK3α/β), total tau and Aβ42 were studied. Results showed that vitamin K2 significantly reduces neuronal cell death by inhibiting cytotoxicity and ROS levels and helps in the retainment of mitochondrial membrane potential. Moreover, vitamin K2 significantly decreased the expression of CHOP protein along with the levels and the nuclear localization of p-IRE1α, thus showing its significant role in inhibiting chronic ER stress-mediated UPR and eventually cell death. In addition, vitamin K2 significantly down-regulated the expression of GSK3α/β together with the levels of total tau protein, with a petite effect on secreted Aβ42 levels. These results suggested that vitamin K2 alleviated mitochondrial damage, ER stress and tauopathy-mediated neuronal cell death, which highlights its role as new antioxidative therapeutics targeting related cellular processes.

## 1. Introduction

Neurodegenerative disease is an umbrella term for a range of neurological disorders that primarily involve the progressive degeneration or death of neurons [1]. The cause of neuronal cell death is implicated in the abnormal folding of proteins inside the brain along with concomitant factors such as neuroinflammation, apoptosis, neurotoxicity and oxidative stress [2]. Among various types, Alzheimer’s disease (AD) is the most common type of neurodegenerative disorders, representing about 60–70% of dementia cases. Statistically, the majority of AD patients (>95%) who develop this disease are >65 years of age [3]. There are approximately 25 million people reported to be affected by dementia globally [4]. Alarmingly, this forecasted number is going to double every 20 years up to 2040, which may lead to an expensive burden of disease on the society in the near future [5]. In both developed and developing nations, Alzheimer’s disease has a severe influence on the affected individuals, caregivers, and society.

Neuronal changes or lesions in AD are characterized by the formation and accumulation of amyloid-beta (Aβ) plaques extracellularly along with the deposition of the intraneuronal hyperphosphorylated tau protein [6]. Besides these two major hallmarks of AD, there are many factors that make its pathogenesis complicated, including mitochondrial quality control, oxidative stress, insulin resistance and autophagy [7]. Accumulated evidence reveals that the multifaceted AD pathogenesis starts with the accumulation of Aβ oligomers, which directly impacts the mitochondrial function via various molecular pathways. This results in decreased mitochondrial membrane potential, uncoupling of mitochondrial electron transport chain, and reduced ATP production, which ultimately leads to mitochondrial dysfunction [8]. The damage to mitochondria results in an imbalance between the production of free radicals and the antioxidative defense mechanism, causing chronic oxidative stress and triggering chronic endoplasmic reticulum (ER) stress [9]. Chronic ER stress is marked by the activation of protein kinase RNA such as ER kinase (PERK), which then activates transcription factor 6 (ATF6), resulting in the stimulation of C/EBP homologous protein-10 (CHOP) that, in turn, increases proapoptotic proteins [10]. During chronic ER stress, inositol-requiring kinase 1α (IRE1α), an evolutionarily conserved transmembrane ER protein that activates tumor necrosis factor receptor-associated factor 2 (TRAF2)-dependent proapoptotic kinases, apoptosis signal-regulating kinase 1 (ASK1), p38, and c-Jun N-terminal kinase (JNK) and causes caspase-mediated apoptosis [11]. Heightened ER stress response also results in increased oxidative stress along with autophagy dysfunction [10]. On the other hand, studies also reveal that oxidative stress is considered the major etiology in AD, which contributes to impairment of mitochondrial function [12], ER stress [13], autophagy dysfunction [14], dysfunctional glucose metabolism [15], and an increase in tauopathy and Aβ secretion [12]. Increasing evidence also shows that elevated levels of glycogen synthase kinase 3 (GSK3), a serine/threonine protein kinase, are found in human and mouse models of AD. Increased expression of GSK3 was found to be correlated with increased levels of phosphorylated tau, Aβ production, and mitochondrial dysfunction [16]. Thus, the interrelation of various pathways involved in the pathogenesis of AD makes it very complicated and incurable by current therapeutic regimens. 

Until now, only five drugs have been approved by the FDA, USA, for the treatment of AD; they include glutamate agonist memantine and acetylcholineesterase (AchE) inhibitors such as rivastigmine, donepezil, and galantamine [17]. These drugs are involved in an improved amyloid-beta precursor protein (APP) processing via inhibition and/or activation of the enzymatic machinery. Other drug regimens are aimed at immunotherapeutic approaches to existing amyloids, tau centric therapies, which include the use of putative tau kinase inhibitors, microtubule stabilizers, and tau immunotherapies [18]. Moreover, therapies to combat oxidative stress are currently being devised because oxidative stress plays a major role in neurodegeneration [19]. Interestingly, current studies are being focused on the screening of bioactive compounds for their effect on the prevention of Aβ formation, oxidative stress reduction and targeting organelle dysfunction, which will potentiate the use of bioactive compounds to prevent AD in early stages [20].

Vitamin K is one of the bioactive compounds that has gained importance in recent years for its role as an antioxidant [21]. As a naturally occurring fat-soluble vitamin, it exists in three biologically active forms: phylloquinone (vitamin K1) and menaquinone (vitamin K2), and one industrial analog used in chemotherapeutic treatments, methylnaphthoquinone (vitamin K3). Though vitamin K1 is abundant in green leafy vegetables, vitamin K2 is found in meat, cheese, soybean, eggs, and curds [22]. Vitamin K2 is produced by gut bacteria and is the most abundant form of vitamin K found in the human brain [23]. Vitamin K2 is found to play an important role in regulating the biosynthesis of sphingolipids, Gas 6 (growth arrest-specific protein 6) signaling and cognitive functions in the brain [24]. Reports showed the role of vitamin K2 in regulating inflammatory cascade activation and in protection from oxidative stress-induced mitochondrial damage in vitro [25]. Moreover, depletion of menaquinone 4 was found to be correlated with poor cognitive performance in the murine model [26]. In addition, it was found that lower intake of vitamin K diet was directly associated with cognitive deficits in the aged population [27]. Current studies also showed the antioxidative effect of vitamin K2 on various in vitro models such as differentiated rat pheochromocytoma PC 12 cells [28] and rat astroglia C6 cells by inhibiting the toxicity and oxidative damage induced by Aβ involving various regulatory pathways [29]. Moreover, vitamin K2 was also observed to significantly inhibit the rotenone-induced ROS production, p38 activation and caspase activation in microglial cells [30]. 

Studies performed till date support the potential of vitamin K2 as a neuroprotective antioxidant molecule that protects the neuron cells in vitro against induced oxidative stress conditions and induced Aβ toxicity. However, the promising role of vitamin K2 as an antioxidant targeting mitochondrial dysfunction, ER stress, and chronic ER stress-mediated unfolded protein response along with its effects on the basal level of total tau protein and Aβ has not been elucidated yet. To deepen our research toward this approach, the present study investigated whether vitamin K2 modulates the signaling of neuronal cell death induced by organelle dysfunction (ER and mitochondria) under toxic conditions generated by streptozotocin and menadione in SH-SY5Y neuroblastoma cells. SH-SY5Y cells were used in this study because of their close resemblance to neurons and are a widely accepted model for neurodegenerative diseases [31,32]. To unveil the molecular mechanism, SH-SY5Y cells were treated with streptozotocin, which is a glucosamine nitrosourea compound used for the experimental simulation of sporadic AD in vivo [33]. Streptozotocin is also known to induce ER stress [34], oxidative stress and mitochondrial dysfunction in vitro [35]. In addition, SH-SY5Y cells were also treated with menadione, a synthetic naphthoquinone, which is known to induce chronic oxidative stress, leading to mitochondrial damage and cell death [36,37]. Our study demonstrates a potential role of vitamin K2 in ameliorating oxidative stress-induced chronic ER stress, unfolded protein response (UPR), associated mitochondrial dysfunction, and GSK3α/β expression. Moreover, this study also unveils the fact that vitamin K2 helps in decreasing the level of total tau protein, which is increased in the presence of streptozotocin with a diminutive effect on the basal level of secreted Aβ42 under stressful conditions.

## 2. Materials and Methods

### 2.1. Mammalian Cell Culture

Human neuroblastoma SH-SY5Y cells were gifted by Dr. Juan Cruz Landoni, Research Program for Molecular Neurology, University of Helsinki. Cells were grown in a T-75 cm^2^ cell culture flask in Dulbecco’s modified Eagle medium (DMEM) (Gibco™ DMEM High Glucose, Thermo Fisher Scientific, Waltham, MA, USA) supplemented with 10% fetal bovine serum (FBS) (Thermo Fisher Scientific, USA) and streptomycin (50 µg/mL)/penicillin (50 U/mL) (Gibco™ Thermo Fisher Scientific, USA) and maintained at 37 °C in a humidified incubator with the supply of 5% CO_2_. For experimental assays, cells were trypsinized by 0.2% trypsin (Thermo Fisher Scientific, USA) and seeded at a density of 5 × 10^4^ in 96-well plates (Costar, Corning, NY, USA). 

### 2.2. Experimental Design for In Vitro Studies

To study the modulating effect of vitamin K2 (4774, Merck, Sigma-Aldrich, Saint Louis, MO, USA) on neuronal toxicity induced by streptozotocin (S0130, Sigma-Aldrich, Saint Louis, USA) and menadione (M5625, Sigma-Aldrich, Saint Louis, MO, USA), time-dependent dose–response experiments were employed for the standardization of vitamin K2 and streptozotocin concentrations. For vitamin K2; 1, 10, 15, 25, 50, and 100 µM and for streptozotocin; 1, 1.5, 2 and 2.5 mM concentrations were used. We obtained the best cell survival rate at 50 µM of vitamin K2, although 2.5 mM for 24 h was the maximum dose for streptozotocin. For the experiments, the following treatment groups were taken: (1) control, (2) vitamin K2, (3) streptozotocin, (4) streptozotocin + vitamin K2, (5) menadione, and (6) menadione + vitaminK2. For each experiment, after 20–24 h of cell passage for proper cell adherence, cells were treated with varied concentrations of streptozotocin (1 mM, 1.5 mM, 2 mM, and 2.5 mM) for 24 h, followed by post-treatment of vitamin K2 at a fixed concentration of 50 µM for 5 h. In case of menadione-induced toxicity, cells were treated with varied concentrations of menadione (100 nM, 250 nM, 500 nM, 1 µM, and 10 µM) for 2 h as standardized previously [36], followed by post-treatment of a fixed concentration of vitamin K2 for 5 h. All experiments were performed in triplicate to quadruplet. All the experiments were performed after treatment with vitamin K2 and all the data presented are blank corrected.

### 2.3. Mitochondrial Activity (MTT) Assay

Mitochondrial activity/cell viability was measured using 3-(4,5-dimethylthiazol-2-thiazolyl)-2,5 diphenyl tetrazolium bromide or thiazolyl blue tetrazolium bromide (MTT; Sigma-Aldrich, St. Louis, MO 63103, USA). Briefly, after the treatment, the medium was removed from the well plates and the assay-specific probe (MTT) was added to 250 µL complete medium (5 mg/mL) and incubated for 3 h at 37 °C. The formazan crystals so formed were dissolved in 250 µL DMSO in each well. Absorbance was measured at a wavelength of 550 nm using a microplate reader (Cytation 3; BioTek Instruments, Inc., Winooski, VT, USA).

### 2.4. Reactive Oxygen Species

ROS production in response to treatment was assayed using 2,7-dichlorofluorescein diacetate (DCFH-DA) (Sigma-Aldrich, St. Louis, MO, USA). Immediately after the treatment, the medium was removed from the wells and replaced with DCF-DA (40 µM) in 0.5 mL of Hank’s balanced salt solution and thereafter incubated for 30 min at 37 °C. Fluorescence was then measured at 485 nm excitation/535 nm emission wavelengths using a BioTek Cytation Reader 3 (BioTek Instruments, Inc., Winooski, VT, USA).

### 2.5. Lactate Dehydrogenase (LDH) Assay

Damage to the plasma membrane releases LDH into the cell culture medium and is indicative of cell cytotoxicity. Cells were seeded at a density of 5 × 10^4^ cells/well in 96-well plates. After the treatment of cells with streptozotocin and menadione followed by post-treatment of vitamin K2, spontaneous LDH release and maximum LDH release were analyzed using a CYQUANT LDH cytotoxicity assay kit (C20300, Invitrogen, Thermo Fisher Scientific, USA). LDH activity was measured spectrophotometrically at 490 nm and 680 nm using a microplate reader (Cytation 3; BioTek Instruments, Inc., Winooski, VT, USA). Total LDH activity was defined as the sum of intracellular and extracellular LDH activities and the amount of LDH released was expressed as a percentage of the total value.

### 2.6. Organelle Dysfunction Studies

#### 2.6.1. Measurement of Mitochondrial Membrane Potential by the JC1 Assay

Mitochondrial apoptotic cascade involves the depolarization of mitochondrial membrane, which can be measured fluorometrically using JC1 dye (JC-1; CAS 47729-63-5; Calbiochem; Sigma-Aldrich, USA). Briefly, 5 × 10^4^ cells were seeded per well in 96-well plates and treated as described in Section 2. After the treatment, cells were incubated with JC1 staining solution (5 µg/mL) at 37 °C for 20 min followed by washing of the cells twice with PBS. Mitochondrial membrane potential was estimated by measuring the fluorescence ratio of JC1 aggregates in mitochondria (red) (λ_ex_ 550 nm, λ_em_ 600 nm) to that of free JC1 monomers (green) (λ_ex_ 485 nm and λ_em_ 535 nm) using a fluorescence plate reader (Biotek Cytation Reader 3; BioTek Instruments, Inc., Winooski, VT, USA). Mitochondrial membrane depolarization is indicated by an increase in the proportion of green JC1 monomers in mitochondria, whereas repolarization of healthy mitochondria is indicated by an increase in red JC1 aggregates.

#### 2.6.2. Detection of Endoplasmic Reticulum Stress Marker (CHOP and p-IRE1α) by Immunofluorescence

To investigate whether streptozotocin and menadione induce ER stress and determine the role of vitamin K2 in attenuating it, expression of the components of ER stress-mediated apoptosis marker C/EBP homologous protein (CHOP), and ER stress sensor of UPR signaling, inositol requiring enzyme 1 (p-IRE1α) was detected by immunofluorescence in SH-SY5Y cells. Briefly, 5 × 10^4^ cells were plated in 8-well chamber slides, and 24 h after seeding, the cells were treated with streptozotocin, and menadione, followed by post-treatment of vitamin K2 as described in Section 2. Consequently, the cells were fixed with 4% paraformaldehyde for 30 min after washing with PBS for 3 times. The fixed cells were incubated with 0.4% Triton X-100 and 4% BSA in PBS for 2 h at room temperature followed by incubation with specific primary antibodies against CHOP (1:200, SAB4500631 Anti-CHOP, Sigma-Aldrich, MO, USA) and p-IRE1α (1:200, 10156403, Thermo Fisher Scientific, USA) at 4 °C overnight. After washing with PBS, the cells were incubated with goat anti-rabbit IgG (1:1000 dilution) (AF-568, Thermo Fisher Scientific, USA) secondary antibody for 1 h at room temperature. All images were obtained using a confocal microscope (LSM 710) using a 63× magnification oil objective, λ_ex_ = 578 nm, λ_em_ = 603 nm). The mean fluorescence intensity from three different fields of view and from three independent experiments was quantified using ImageJ [38].

### 2.7. Detection of GSK3α/β by Immunofluorescence

GSK3α/β is a constitutively active serine/threonine kinase regulated by phosphorylation at serine /threonine residues and is found to play an important role in mediating ER stress-induced apoptosis, affecting CHOP expression and tau hyperphosphorylation [39]. To investigate the role of vitamin K2 in attenuating the effect of ER stress-induced neuronal cell death, expression of the active form of GSK3α/β was detected by immunofluorescence using a GSK-3 alpha/beta [p-Tyr216, p-Tyr279] (Novus Biologicals, USA) primary antibody. The procedure followed here was the same as in Section 2.7. Anti-GSK3 alpha/beta [p-Tyr216, p-Tyr279]) was diluted 1:200 and detected by goat anti-rabbit IgG (AF488, Thermo Fisher Scientific, USA). All images were obtained using a confocal microscope (LSM 710) using a 63× magnification oil objective, λ_ex_ = 490 nm, λ_em_ = 525 nm). The mean fluorescence intensity from three different fields of view and from three independent experiments was quantified using ImageJ [38].

### 2.8. Detection of Neurodegenerative Marker Proteins 

#### Detection of Human (Total) Tau and Aβ42 Levels by ELISA

Alteration in the levels of intracellular total tau protein and extracellular Aβ42 in SH-SY5Y cells under the toxic conditions (streptozotocin and menadione treatment) and subsequent treatment with vitamin K2 was detected using the tau (total) human ELISA kit (KHB 0041, Thermo Fisher Scientific, USA) and the Aβ42 human ELISA kit (KHB 3441, Thermo Fisher Scientific, USA). Briefly, cells were seeded at a density of 5 × 10^4^ cells in 96-well plates. After the treatment, the supernatant was collected for estimation of secreted Aβ42 and then cells were lysed for total tau protein estimation. The lysates were centrifuged at 1300 rpm for 10 min at 4 °C to remove any cell debris, and the protein concentration was estimated according to the manufacturer’s instructions.

### 2.9. Statistical Analysis

All data are presented as mean ± SEM and were compared with the respective control groups. Statistical analysis between two groups (for example, streptozotocin and its post-treatment with vitamin K2) was performed using Student’s *t*-test utilizing GraphPad Prism 9, and differences with *p* ≤ 0.05 were designated as significant in all the cases. * *p* < 0.05, ** *p* < 0.01, and *** *p* < 0.001 denote significant differences compared to the control and between the treatment groups.

## 3. Results

### 3.1. Vitamin K2 Protects the SH-SY5Y Cells from Streptozotocin- and Menadione-Induced Decrease in Cell Viability

The effect of streptozotocin and menadione on the viability of SH-SY5Y neuroblastoma cells was first examined using the MTT assay. It was observed that streptozotocin (1, 1.5, 2, and 2.5 mM) significantly reduced the number of viable cells in a dose-dependent manner when compared to non-treated cells taken as control. Similarly, menadione at concentration (100 nM, 250 nM, 500 nM, 1 µM, and 10 µM) reduced the number of viable cells within 2 h of treatment in a dose-dependent manner compared to control cells, whereas the cells treated with only vitamin K2 were found to be non-toxic. Subsequently, in both the cases (streptozotocin and menadione treatment), post-treatment of vitamin K2 for 5 h resulted in the recovery of damaged cells, which shows that vitamin K2 significantly inhibited the toxic effects of increasing concentrations of streptozotocin and menadione as shown in Figure 1A,B.

### 3.2. Vitamin K2 Reduces Cell Cytotoxicity Induced by Streptozotocin and Menadione

The effect of streptozotocin and menadione on the integrity of the cell membrane was further confirmed by an LDH release test. When SH-SY5Y cells were treated with streptozotocin in a dose-dependent manner for 24 h, it was observed that streptozotocin significantly increased the percentage cell cytotoxicity with an increasing concentration, i.e., a high LDH release was observed at a concentration of 2.5 mM. When cells treated with streptozotocin were exposed to vitamin K2 for 5 h (post-treatment), a significant reduction in cytotoxicity was observed, which indicates that vitamin K2 helped the cells to regain their metabolic activity (Figure 2A). In a similar way, when SH-SY5Y cells were treated with menadione for 2 h, membrane integrity was found to be significantly compromised, which was found to be significantly improved by post-treatment of vitamin K2 for 5 h (Figure 2B). This assay confirmed the cytoprotective role of vitamin K2 even at a maximum dose of streptozotocin and menadione.

### 3.3. Vitamin K2 Attenuated ROS Production Induced by Streptozotocin and Menadione

Treatment of cells with streptozotocin at varied concentrations (1, 1.5, 2 and 2.5 mM) increased the intracellular ROS accumulation, which was indicated by the significant increase of DCFH-DA fluorescence compared to control and only vitamin K2-treated cells. While post-treatment of streptozotocin-treated cells with vitamin K2 significantly reduced the ROS-dependent fluorescent signal even at higher concentration of streptozotocin-treated cells (Figure 3A). Similarly, in cells treated with menadione (100 nM, 250 nM, 500 nM, 1 µM, and 10 µM), ROS was found to be significantly increased when compared to the control and only vitamin K2-treated cells, while post-treatment of menadione-treated cells with vitamin K2 for 5 h led to the significant reduction in DCFH-DA fluorescence, indicating a significant reduction in ROS (Figure 3B). 

### 3.4. Vitamin K2 Inhibited Mitochondrial Depolarization Induced by Streptozotocin and Menadione

Loss of mitochondrial membrane potential is indicative of mitochondrial uncoupling and is symbolic of early signs of apoptosis. We determined the mitochondrial membrane potential in SH-SY5Y cells by the shift in JC-1 fluorescence from red to green, where red fluorescence indicates the formation of JC-1 aggregates in healthy mitochondria and green fluorescence indicates the JC-1 monomers or loss of membrane potential. When cells were exposed to various concentrations of streptozotocin, it resulted in significant membrane potential dissipation, as observed by the high shift toward green fluorescence and subsequently a lower red/green ratio compared to the control and cells treated with only vitamin K2 (Figure 4A). While post-treatment of vitamin K2 for 5 h resulted in a significant retainment of mitochondrial membrane potential as indicated by a significantly high red/green ratio (Figure 4A). Likewise, loss of mitochondrial membrane potential (significantly higher shift from red to green fluorescence) was observed in case of menadione treatment of the cells and post-treatment with vitamin K2 helped in retainment of mitochondrial functionality even at a high dose of menadione (Figure 4B). 

### 3.5. Vitamin K2 Reduced Endoplasmic Reticulum Stress-Induced by Streptozotocin and Menadione

Protein expression of one of the major ER stress-mediated apoptotic pathway proteins CHOP (C/EBP homologous protein, also known as growth arrest and DNA damage-inducible gene 153 (GADD153)) was studied by immunofluorescence in streptozotocin- and menadione-treated cells at their highest toxic concentrations, as well as in post-treatment of cells with vitamin K2. From confocal microscopy images (Figure 5A) along with the mean fluorescence intensity quantified using ImageJ (Figure 5B), it was observed that treatment of cells with streptozotocin (2.5 mM) causes augmented expression of CHOP in comparison with that of the control and only vitamin K2-treated cells. While post-treatment of vitamin K2 for 5 h in streptozotocin-treated cells resulted in a significant decrease in the expression of CHOP. In case of menadione-treated cells, a significant increase in the expression of CHOP was observed, which indicated that the menadione-induced ER stress-mediated apoptosis in cells and vitamin K2 post-treatment resulted in a significant decrease in the expression of CHOP. No significant expression of CHOP was observed in the case of only VK2-treated cells. This shows that vitamin K2 availed the balance between survival signaling and apoptosis signaling.

### 3.6. Vitamin K2 Reduced Endoplasmic Reticulum Stress-Induced UPR Signaling Mediated by Streptozotocin and Menadione

Exogenous factors such as streptozotocin and menadione resulted in alterations in ER homeostasis, stimulating the activation of unfolded protein response (UPR). UPR acts as an adaptive pathway taken up by the cells to re-establish the ER homeostasis or prolonged ER stress might lead to the activation of the UPR-mediated proapoptotic pathway. To observe the modulating effect of vitamin K2 on UPR signaling pathways, the expression of UPR, inositol-requiring enzyme (IRE1), was investigated by confocal microscopy. The C-terminal domain of IRE1 is activated via autophosphorylation in response to ER stress, forms higher order oligomers, and is involved in mediating pro-survival or proapoptotic pathways [40]. In streptozotocin- and menadione-treated cells, p-IRE1α was found to be highly expressed in comparison with that of the control and only vitamin K2-treated cells. Moreover, p-IRE1α was also found to be translocated to the nucleus in case of streptozotocin and menadione treatment, which indicated the induction of UPR response and the role of p-IRE1α in splicing (Figure 6A). Whereas post-treatment of vitamin K2 in streptozotocin- and menadione-treated cells resulted in reduced expression of p-IRE1α and no nuclear translocation was observed, similar to that of the control and only vitamin K2-treated cells. Similarly, the reduction in mean fluorescence intensity observed in case of vitamin K2 treatment (Figure 6B) further confirms its role in reducing the expression of p-IRE1α, and thus ER stress-mediated UPR signaling. 

### 3.7. Vitamin K2 Reduced the Expression of p GSK3α/β Induced by Streptozotocin and Menadione

Due to the involvement of GSK3α/β in different signal transduction pathways including induction of mitochondrial dysfunction, tau hyperphosphorylation and its role in ER stress-induced CHOP, expression of the active form of GSK3 (GSK3α/β (p-Tyr216, p-Tyr279)) was observed by confocal microscopy and the mean fluorescence intensity was calculated using ImageJ [38]. In streptozotocin- and menadione-treated cells, p-GSK3α/β was found to be highly expressed in comparison with that of the control and only vitamin K2-treated cells (Figure 7A,B); this indicated the role of the streptozotocin- and menadione-induced GSK3-mediated apoptotic pathway and might be its subsequent role in mitochondrial dysfunction. By contrast, post-treatment of vitamin K2 in streptozotocin- and menadione-treated cells resulted in the subsequent reduction in the expression of active GSK3α/β [p-Tyr216, p-Tyr279], similar to that of the control and only vitamin K2-treated cells (Figure 7A,B).

### 3.8. Effect of Vitamin K2 on Intracellular Total Tau Protein Expression

One of the mechanisms underlying Alzheimer’s disease is the deposition of the neurofibrillary tangles of the microtubule binding protein tau. About 20 different neurodegenerative diseases are directly linked to “tauopathies” [41]. Therefore, we determined the alterations in the level of intracellular total tau protein (irrespective of their phosphorylation state) by ELISA in SH-SY5Y cells in response to streptozotocin and menadione treatment followed by the post-treatment of vitamin K2. A best-fit standard curve was plotted for the known protein concentrations (0–2000 pg/mL) using graph pad prism. Results showed that treatment of cells with streptozotocin at various concentrations significantly increased the levels of intracellular total tau protein (STZ 1 mM—2572.64 pg/mL, STZ 1.5 mM—2413.55 pg/mL, STZ 2 mM—2436.27 pg/mL, STZ 2.5 mM—2318 pg/mL) in comparison to that of control (2245.36 pg/mL) and only VK2 treated cells (2159 pg/mL). On the other hand, post-treatment of streptozotocin-treated cells by vitamin K2 for 5h (represented as SV in the graph) led to the significant reduction in tau protein levels (SV 1 mM—2245.36 pg/mL, SV 1.5 mM—2145.36 pg/mL, SV 2 mM—2199.91 pg/mL, SV 2.5 mM—2018.09 pg/mL) (Figure 8A). By contrast, menadione treatment did not significantly alter the total intracellular tau levels (MQ 500 µM—2190.82 pg/mL, MQ 1 mM— 2063.55pg/mL, MQ 10 mM—2240.82 pg/mL, MQ 15 mM—2240.82 pg/mL) in comparison with that of the control (2204.45 pg/mL) and only vitamin K2-treated cells (2277.18 pg/mL), whereas post-treatment with vitamin K2 for 5 h led to the significant decrease of tau levels at high concentration of menadione (MV 500 µM- 2113.55 pg/mL, MQ 1 mM—2090.82 pg/mL, MQ 10 mM—2059 pg/mL, MQ 15 mM—1931.73 pg/mL) (Figure 8B). 

### 3.9. Effect of Vitamin K2 on Secreted Aβ42 

Alterations in the level of secreted Aβ42 were detected by ELISA in SH-SY5Y cells in response to streptozotocin and menadione treatment followed by the post-treatment of vitamin K2. A best-fit standard curve was plotted for the known protein concentrations (0–1000 pg/mL) using Graphpad Prism. The level of secreted Aβ42 was found to be unaltered in response to streptozotocin and menadione treatment in comparison with that of the control (9.33 pg/mL) and only vitamin K2-treated cells (Figure 9A,B). Moreover, post-treatment of cells with vitamin K2 did not affect the levels of secreted Aβ42 in menadione-treated cells but was found to cause a slight decrease in Aβ42 secretion in streptozotocin-treated cells (SV 2–6.8 pg/mL) (Figure 9A,B).

## 4. Discussion

Although the etiology and pathogenesis of AD are not fully understood, the damage caused by oxidative stress is considered one of the prodromal factors involved in the pathophysiology of Alzheimer’s disease. A growing line of evidence links the oxidative stress with impairments in mitochondrial dynamics and function, endoplasmic reticulum homeostasis, autophagic processes, and glucose metabolism, which ultimately leads to cognitive impairment, synaptic dysfunction, and neuronal cell death. In this scenario, the therapy should include the antioxidant which might target mitochondrial dysfunction, ER stress, ER stress–UPR-mediated apoptotic pathway and the related cellular processes. Toward this approach, our study demonstrated the potential role of vitamin K2 in ameliorating the high oxidative stress-induced mitochondrial dysfunction and ER stress along with its ability to significantly reduce the levels of intracellular total tau protein levels and, to a small extent, cause a decrease in Aβ42 levels in human neuroblastoma cells.

To evaluate the ability of vitamin K2 to modulate oxidative stress, mitochondrial dysfunction, ER homeostasis, and proteinopathies related to AD, human neuroblastoma SH-SY5Y cells were exposed to toxic compounds such as streptozotocin and menadione in a dose- and time-dependent manner. Streptozotocin is a highly toxic compound that has been used not only for inducing type 1 diabetes in rodents [42] but can also stimulate sporadic AD in vivo [33]. On the other hand, menadione, a synthetic precursor of vitamin K2, is well known to cause DNA damage by inducing high reactive oxygen species generation (ROS) in vitro [36,43]. The MTT assay quantifies the mitochondrial activity. A critical step in many apoptotic pathways involves the release of apoptotic inducible factors and cytochrome C from damaged mitochondria, which in turn induces the caspase activation and eventually leads to apoptotic cell death [44,45]. Measurement of cell viability or mitochondrial activity revealed that streptozotocin and menadione treatment resulted in a significant decrease in cell viability in comparison with that of the control and only vitamin K2-treated cells. By contrast, post-treatment of vitamin K2 (50 µM) for 5 h has shown a significant protective effect against the toxicity induced by streptozotocin and menadione, and resulted in a significant increase in the viability of cells. These results suggest that vitamin K2 acts to preserve mitochondrial function in streptozotocin- and menadione-treated cells, also indicating its potent antioxidant activity, and might be helpful to elevate the endogenous antioxidant capacity of the cells. Moreover, the LDH assay showed that vitamin K2 significantly improved the cell membrane integrity which was found to be compromised by the treatment of streptozotocin and menadione. Such a neuroprotective role of vitamin K2 was also observed when differentiated rat pheochromocytoma PC12 cells were pre-treated with vitamin K2 and exposed to Aβ (1–42) peptide and H_2_O_2_ [28]. A similar cytoprotective effect of vitamin K2 was observed in astroglioma C6 cells, where it has been found to protect the C6 cells from Aβ-induced toxicity in a dose-dependent manner [29]. Even it was observed that vitamin K2 reduces the toxic effects of rotenone in SH-SY5Y cells, when BV2 cell conditioned medium (rotenone + MK-4) was added to SH-SY5Y cells, thus protecting from microglia-mediated neuronal cell death [30]. Besides the neuroprotective role, vitamin K2 was also found to restore osteoblast function in vitro and in vivo [46] and protect rat smooth muscle cells from apoptosis [47].

Exogenous and some environmental factors or a natural aging process could cause an imbalance in oxidoredox homeostasis, resulting in the increase of ROS and RNS [48]. Overproduction of ROS leads to oxidative stress conditions, which might cause metabolic changes, damage to mt-DNA or genomic instability, ultimately leading to cellular damage and apoptosis [49]. To ameliorate the oxidative stress condition, endogenous and exogenous antioxidants play a very important role. In this study, it was observed that streptozotocin and menadione resulted in significantly increased production of ROS in SH-SY5Y cells when treated in a dose-dependent manner in comparison with that of the control and only vitamin K2-treated cells. While post-treatment of vitamin K2 significantly reduced the ROS levels as indicated by the decreased fluorescence intensity of DCFH-DA, suggesting that vitamin K2 protected against streptozotocin- and menadione-induced toxicity by suppressing ROS accumulation. Similarly, vitamin K2 was found to be effective against Aβ _(1–42)_ [29] and H_2_O_2_ [28] induced ROS generation. Increasing evidence reveals that increased ROS production is associated with damage to mitochondrial DNA and a decrease in mitochondrial membrane potential. This results in a caspase-mediated apoptotic pathway and is considered the principal causative factor of AD pathogenesis [50]. In this study, we found that streptozotocin and menadione resulted in a significant decrease in mitochondrial membrane potential as indicated by the decrease in red to green fluorescence intensity observed by the JC1 assay, thus indicating mitochondrial dysfunction. Streptozotocin was also found to induce mitochondrial dysfunction in various cell types such as HepG2 [35] and pancreatic β [51] cells and also in vivo models [52]. We found that post-treatment of vitamin K2 in streptozotocin- and menadione-treated cells resulted in recovering of mitochondrial membrane potential as indicated by the increased red to green fluorescence intensity of JC1 dye, indicating healthy mitochondria. Thus, vitamin K2 played an important role in preserving mitochondrial function and thus, prevented the activation of mitochondrial dysfunction-related neuronal cell death. Protection of mitochondrial function by vitamin K2 is also supported by a recent report where vitamin K2 was found to increase ATP production and increase the expression of autophagy-related genes in the AD model of *Drosophila* [53]. Moreover, it was observed that vitamin K2 (MK-7) increases ATP levels in patients with Parkinson’s disease and mitochondrial dysfunction [54].

Chronic oxidative stress-mediated mitochondrial dysfunction further marks the accumulation of unfolded proteins in the endoplasmic reticulum, leading to ER stress. To support the survival of ER-stressed cells, an adaptive mechanism unfolded protein response (UPR) is induced in cells, which promotes the removal of misfolded proteins along with the up-regulation of antioxidant factors [9]. However, when the cytoprotective effect of UPR is not achievable and it is severe and prolonged, UPR-mediated proapoptotic response activates specific receptor proteins leading to apoptosis [14]. Three ER membrane localized sensors IRE1α, PERK, and activating transcription factor 6 (ATF6) are involved in mediating UPR, which can determine cell fate by inducing different interconnected downstream signaling cascades [9]. Among the stress-responsive genes activated by the protein kinase RNA such as ER kinase (PERK)- and ATF6-mediated pathway is C/EBP homologous protein 10 (CHOP), which is the factor responsible for the ER stress-mediated apoptosis pathway [10]. During mild ER stress, CHOP expression is not prolonged because it is a short-lived protein. On the contrary, when CHOP expression is persistent and causes cell death, it indicates chronic ER stress [55]. Thus, to study the protective effects of vitamin K2 in response to ER stress, we determined the expression of CHOP in SH-SY5Y cells by immunocytochemistry. It was observed that treatment of SH-SY5Y cells with streptozotocin and menadione significantly increased the expression of CHOP in comparison with that of the control and vitamin K2-treated cells. By contrast, post-treatment of vitamin K2 resulted in a significant reduction in expression of CHOP, suggesting that vitamin K2 helped the cells to cope up with severe ER stress conditions induced by streptozotocin and menadione, and thus vitamin K2 helped to reduce neuronal cell death by a significant reduction in the expression of CHOP. In parallel with this, during prolonged ER stress, inositol-requiring kinase 1α (IRE1α) also mediates apoptosis via the TRAF2-ASK1-p38-JNK pathway [11]. We also determined the expression of p-IRE1α by immunocytochemistry, in the presence of vitamin K2. Interestingly, it was observed that streptozotocin and menadione treatment of cells not only augmented the expression but also resulted in the nuclear translocation of p-IRE1α, which is indicative of the IRE1 proteolysis in SHSY5Y cells and its involvement in splicing. Previous reports also showed that induction of UPR leads to proteolytic cleavage and nuclear translocation of the cytosolic domain of p-IRE1 [56], where presenilin 1 is involved in proteolytic step [40]. By contrast, vitamin K2 treatment in streptozotocin- and menadione-treated cells was found to reduce the expression of p-IRE1α as well as its nuclear translocation. Thus, reduction in the expression of p-IRE1α along with CHOP by vitamin K2 suggests its potential involvement in attenuating chronic ER stress in SH-SY5Y cells. Several studies done so far revealed that high expressions of ER stress markers (GRP78, PERK, p-eIF2α, IRE1α, PDI, ATF4, and CHOP10) are found in the AD brain [57,58]. Moreover, it was observed that the Braak stage in the AD brain was found to be directly associated with upregulated expression of proapoptotic UPR transcription factor CHOP and p IRE1 [59].

Glycogen synthase kinase 3 (GSK3) also plays an important role in executing ER stress-mediated apoptosis as studied using GSK3 inhibitors in a variety of cell types and ER stress models [29,60]. The phosphotyrosine residue of GSK3 is the “activation loop” of this enzyme (Tyr279 in GSK3*α* and Tyr216 in GSK3*β*) and is also linked to activation of MAPK. By contrast, phosphorylation at its serine residue or mutation of a tyrosine residue by phenylalanine inhibits the activity of GSK3 [61]. In the present study, immunocytochemistry of (GSK-3 α/β [p-Tyr216, p-Tyr279]) revealed its very high expression in streptozotocin and menadione treatment at their high doses in comparison with that of the control and only vitamin K2-treated cells, whereas post-treatment of vitamin K2 in streptozotocin- and menadione-treated cells resulted in decreased expression of the active form of GSK3. These results suggest that vitamin K2 assisted in attenuating neuronal cell death by preventing the activation of GSK3α/β. This conclusion was further corroborated by a study where another bioactive compound curcumin was found to inhibit the activity of GSK3β and regulated autophagy in SH-SY5Y cells [62]. In addition, many natural and synthetic bioactive inhibitors of GSK3 are known to reduce the pathophysiology of neurodegenerative diseases [39]. A previous report has also revealed the potential link of GSK3-mediated regulation of ER stress-induced CHOP expression and apoptotic signaling and found that it might be a mechanism selective to cells of neuronal lineage [63]. Moreover, a previous report has also revealed that elevated levels of GSK3β in AD-affected tissues activate VDAC1 phosphorylation in mitochondria [64] causing mitochondrial dysfunction, and increased Aβ production and hyperphosphorylation of tau leading to synaptic damage [16]. 

Together with Aβ42 and p-tau (phosphorylated tau), total tau (t-tau) is considered one of the three core biomarkers of AD [65]. An elevated level of total tau was first reported in AD patients in the year 1995 [66]; since then, with the same reproducible finding in many reports, total tau is considered one of the CSF and blood biomarkers of AD [65]. In the present study, determination of intracellular total tau protein and secreted Aβ42 by ELISA revealed very interesting outcomes. In case of intracellular total tau protein estimation, streptozotocin was found to increase the level of total tau protein in comparison with that of the control and only vitamin K2-treated cells. On the other hand, menadione did not cause a significant increase in total tau protein concentration when compared to the control and only vitamin K2-treated cells. But in both cases, post-treatment of vitamin K2 resulted in a significant decrease in the levels of intracellular total tau protein even at a higher dose of streptozotocin and menadione. In case of secreted Aβ42 levels, both streptozotocin and menadione did not affect the basal level of Aβ42. Indeed, treatment of vitamin K2 somehow decreased the basal levels of Aβ42 in the case of streptozotocin but was found to have no significant effect in the case of menadione. Thus, our study showed the potential role of vitamin K2 in decreasing the levels of streptozotocin-induced intracellular total tau protein. Meta-analysis study done in the recent past indicated the use of cerebrospinal fluid (CSF) t-tau along with p-tau and Aβ42 in the early detection of AD in patients with mid-cognitive impairment(MCI) with a sensitivity of 95% [67]. In accordance with the present study, a recent report showed that vitamin K2 along with inhibition of Nauk1 attenuated sevoflurane-induced tau phosphorylation and cognitive impairment in neonatal mice [68]. Vitamin K2 was also found to be effective against Aβ42-induced toxicity and increased ATP production in Alzheimer’s disease *Drosophila* [53]. In case of AD, activation of p38 MAPK is linked to phosphorylation of tau and the formation of tau tangles leading to apoptosis [69]. In a study, it was found that vitamin K2 helped in reducing the levels of phosphorylated p38 MAPK, thus reducing SH-SY5Y cell death induced by H_2_O_2_ and Aβ42 [28]. Increasing evidence also showed that not only vitamin K2 but also some other fat-soluble vitamins are known to affect the pathophysiology of Alzheimer’s disease [70].

## 5. Conclusions

Taken together, this study demonstrates that streptozotocin and menadione led to an increase in oxidative stress by increasing the levels of reactive oxygen species, which in turn triggered the mitochondrial depolarization leading to mitochondrial dysfunction and further activated the endoplasmic reticulum stress. Interestingly, streptozotocin and menadione were found to increase the GSK3 levels, which is linked to the apoptotic cell death in AD. In concert, treatment of streptozotocin also increased the levels of intracellular total tau protein while menadione was found to be neutral. Nevertheless, these pathological responses were inhibited by the post-treatment of vitamin K2. This showed that vitamin K2 modulated neuronal cell death not only by improving mitochondrial health but alsoby inhibiting ER stress, decreasing the levels of GSK3, and inhibiting the increase in the level of total tau protein, as summarized in Figure 10. Thus, from our study, it can be concluded that vitamin K2 could be considered a preventive novel antioxidative therapeutics for AD, which is able to target the impairment of mitochondrial activities, ER stress, ER stress-mediated UPR, and tauopathy. Further research is still needed to elucidate the detailed mechanisms through which vitamin K2 affects ER stress and autophagic signaling to better understand its neuroprotective role.

## Figures and Tables

**Figure 1 antioxidants-10-00983-f001:**
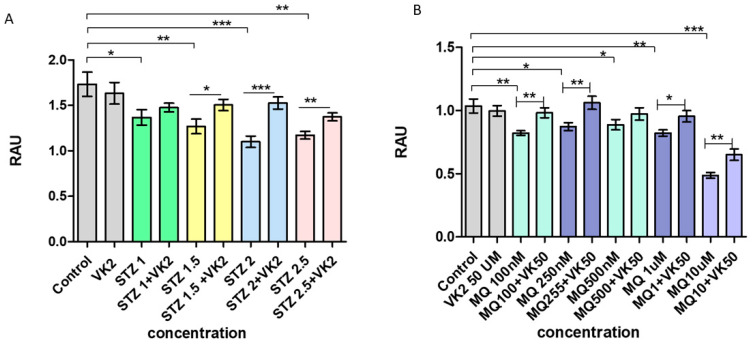
Measurement of cell viability in human SH-SY5Y neuroblastoma cells after (**A**) streptozotocin treatment and post-treatment with vitamin K2 (50 µM) and (**B**) menadione treatment (MQ) and post-treatment with vitamin K2. The *Y*-axis represents the relative absorbance unit (RAU) at 550 nm and the *X*-axis represents the concentration of streptozotocin and menadione. The data are presented as mean ± SEM. Statistical analysis was performed using Student’s *t*-test and differences with *p*-values ≤ 0.05 were designated as significant. * *p* < 0.05, ** *p* < 0.01, and *** *p* < 0.001 denote significant differences compared to the control and between the treatment groups. (STZ—streptozotocin; VK2—vitamin K2; SV—post-treatment of streptozotocin (1–2.5 mM)-treated cells with vitamin K2 (50 µM); MQ—menadione; MQ + VK2—post-treatment of menadione (100 nM–10 µM)-treated cells with vitamin K2 (50 µM).).

**Figure 2 antioxidants-10-00983-f002:**
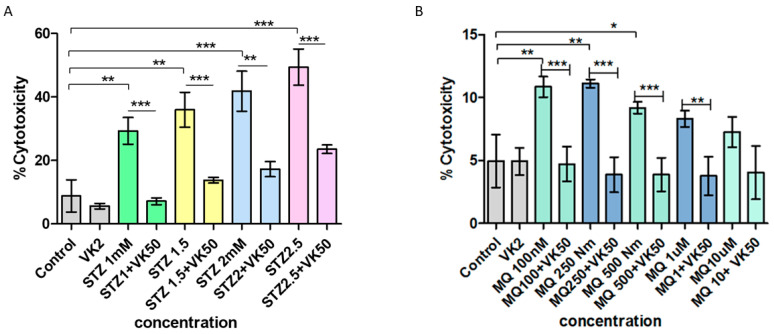
Measurement of percentage cell cytotoxicity using the LDH release assay in human SH-SY5Y neuroblastoma cells after (**A**) streptozotocin treatment and post-treatment with vitamin K2 (50 µM) and (**B**) menadione treatment (MQ) and post-treatment with vitamin K2. The *Y*-axis represents the percentage cytotoxicity and the *X*-axis represents the concentrations of streptozotocin and menadione. The data are presented as the mean ± SEM. Statistical analysis was performed using Student’s *t*-test and differences with *p*-values ≤ 0.05 were designated as significant. * *p* < 0.05, ** *p* < 0.01, and *** *p* < 0.001 denote significant differences compared to the control and between the treatment groups. (STZ—streptozotocin; VK2—vitamin K2; SV—post-treatment of streptozotocin (1–2.5 mM)-treated cells with vitamin K2 (50 µM); MQ—Menadione; MQ + VK2—post-treatment of menadione (100 nM–10 µM)-treated cells with vitamin K2 (50 µM).).

**Figure 3 antioxidants-10-00983-f003:**
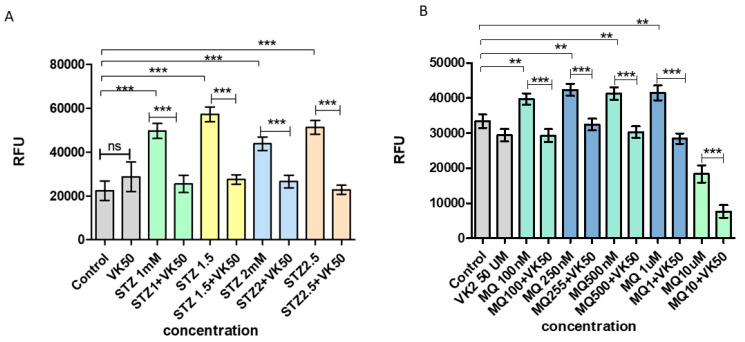
Measurement of reactive oxygen species production in human SH-SY5Y neuroblastoma cells after (**A**) streptozotocin treatment and post-treatment with vitamin K2 (50 µM) and (**B**) menadione treatment (MQ) and post-treatment with vitamin K2. The *Y*-axis represents the relative fluorescence unit (RFU) and the *X*-axis represents the concentrations of streptozotocin and menadione. The data are presented as mean ± SEM. Statistical analysis was performed using Student’s *t*-test and differences with *p*-values ≤ 0.05 were designated as significant. ** *p* < 0.01, and *** *p* < 0.001 denote significant differences compared to the control and between the treatment groups. (STZ—streptozotocin; VK2—vitamin K2; SV—post-treatment of streptozotocin (1–2.5 mM)-treated cells with vitamin K2 (50 µM); MQ-menadione; MQ + VK2—post-treatment of menadione (100 nM–10 µM)-treated cells with vitamin K2 (50 µM).).

**Figure 4 antioxidants-10-00983-f004:**
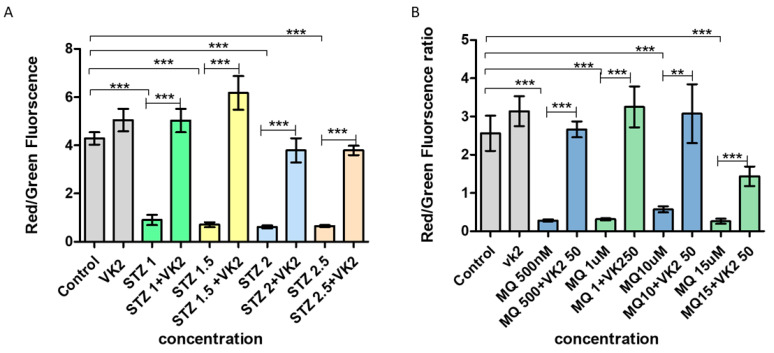
Measurement of mitochondrial membrane depolarization state using the JC1 assay in human SH-SY5Y neuroblastoma cells after (**A**) streptozotocin treatment and post-treatment with vitamin K2 (50 µM) and (**B**) menadione treatment (MQ) and post-treatment with vitamin K2. The *Y*-axis represents the red/green fluorescence ratio and the *X*-axis represents the concentrations of streptozotocin and menadione. The data are presented as mean ± SEM. Statistical analysis was performed using Student’s *t*-test and differences with *p*-values ≤ 0.05 were designated as significant. ** *p* < 0.01, and *** *p* < 0.001 denote significant differences compared to the control and between the treatment groups. [STZ—streptozotocin; VK2—vitamin K2; SV—post-treatment of streptozotocin (1–2.5 mM)-treated cells with vitamin K2 (50 µM); MQ—menadione; MQ + VK2—post-treatment of menadione (500 mM–15 µM)-treated cells with vitamin K2 (50 µM)].

**Figure 5 antioxidants-10-00983-f005:**
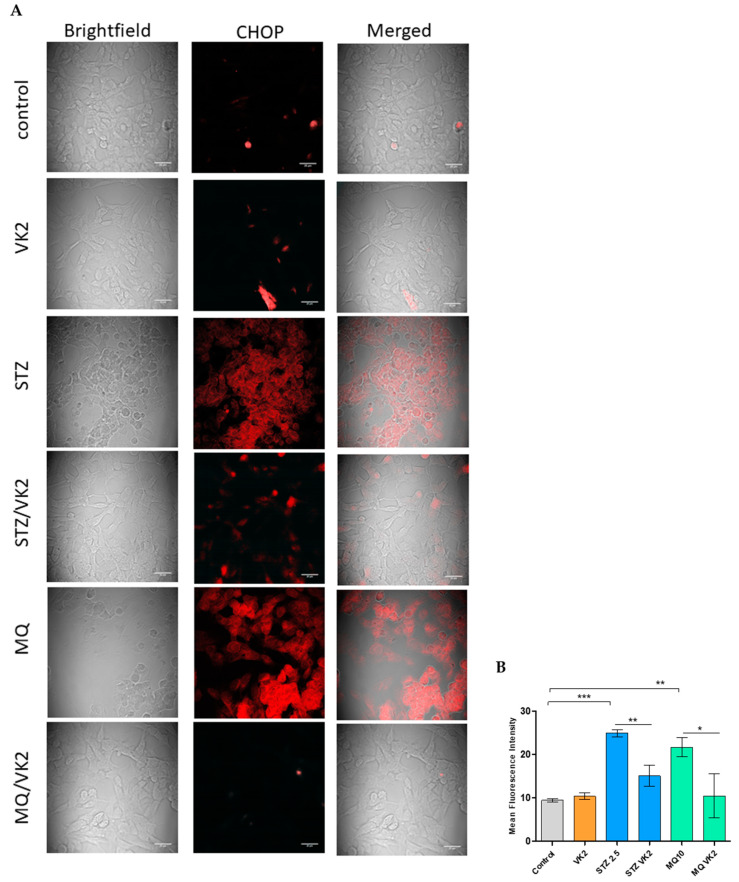
Determination of CHOP expression by immunofluorescence staining using confocal microscopy. (**A**) Images of human SH-SY5Y neuroblastoma cells were taken after streptozotocin and menadione (MQ) exposure and post-treatment with vitamin K2. All images were taken with a 63× oil objective. Images were taken in brightfield and Texas red modes, where AF568 (red staining) represents CHOP. Further merged images were used to observe the differences among the control, only VK2, streptozotocin (2.5 mM), and its post-treatment with vitamin K2 (50 µM); menadione (10 µM) and its post-treatment with vitamin K2 (50 µM). (**B**) Representative graphs of the mean fluorescence intensity from three different fields of view using ImageJ software. The data are presented as mean ± SEM. Statistical analysis was performed using Student’s *t*-test and differences with *p*-values ≤ 0.05 were designated as significant. * *p* < 0.05, ** *p* < 0.01, and *** *p* < 0.001 denote significant differences compared to the control and between the treatment groups. [STZ—streptozotocin; VK2—vitamin K2; STZ/VK2—post-treatment of streptozotocin (2.5 mM)-treated cells with vitamin K2 (50 µM); MQ—menadione; MQ/VK2—post-treatment of menadione (10 µM)-treated cells with vitamin K2 (50 µM)].

**Figure 6 antioxidants-10-00983-f006:**
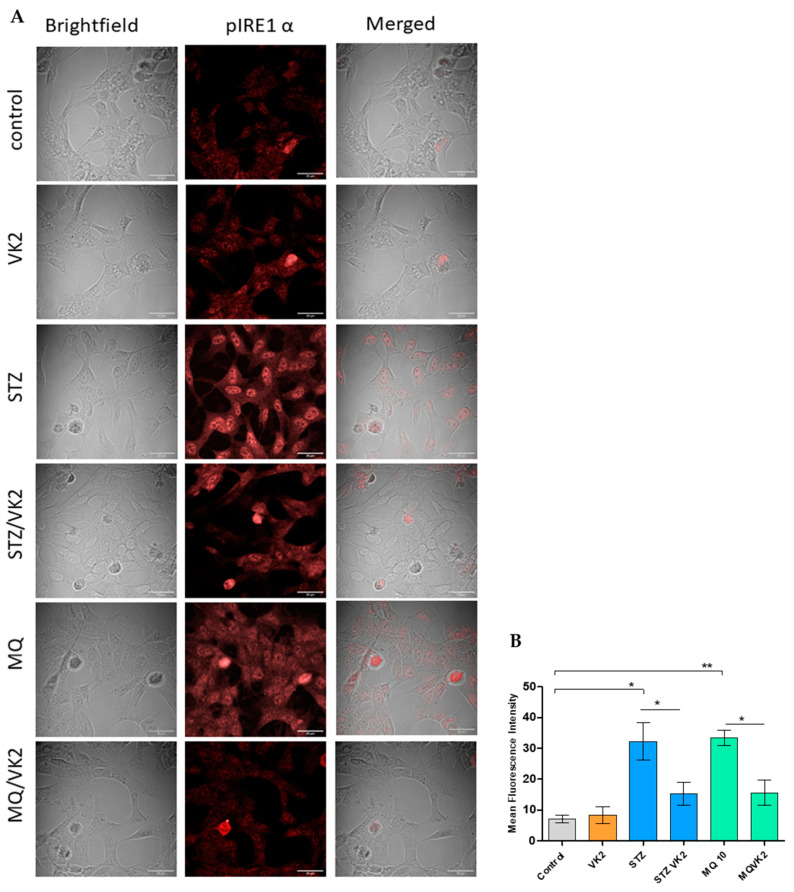
Determination of p-IRE1α expression by immunofluorescence staining using confocal microscopy. (**A**) Images of human SH-SY5Y neuroblastoma cells were taken after streptozotocin and menadione (MQ) exposure and post-treatment with vitamin K2. All images were taken with a 63× oil objective. Images were taken in brightfield and Texas red modes, where AF568 (red staining) represents p-IRE1α. Further merged mages were used to observe the differences among the control, only VK2, streptozotocin (2.5 mM) and its post-treatment with vitamin K2 (50 µM); menadione (10 µM) and its post-treatment with vitamin K2 (50 µM). (**B**) Representative graphs of the mean fluorescence intensity from three different fields of view using ImageJ software. The data are presented as mean ± SEM. Statistical analysis was performed using Student’s *t*-test and differences with *p*-value ≤ 0.05 were designated as significant. * *p* < 0.05 and ** *p* < 0.01 denote significant differences compared to the control and between the treatment groups. [STZ—streptozotocin; VK2—vitamin K2; STZ/VK2—post-treatment of streptozotocin (2.5 mM)-treated cells with vitamin K2 (50 µM); MQ—enamdione; MQ/VK2— post-treatment of menadione (10 µM)-treated cells with vitamin K2 (50 µM)].

**Figure 7 antioxidants-10-00983-f007:**
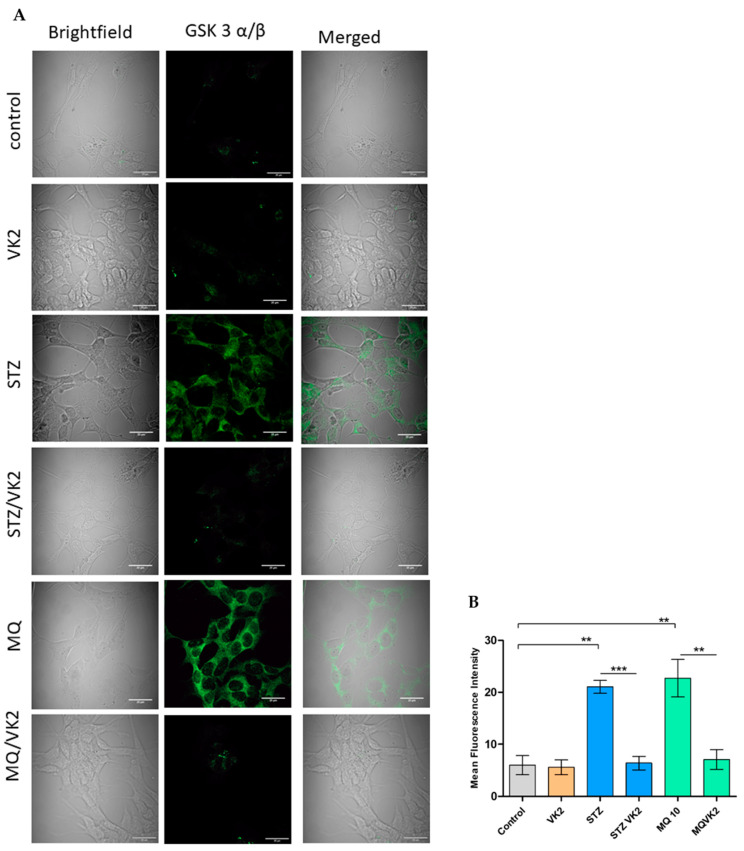
Determination of GSK-3 alpha/beta (p-Tyr216, p-Tyr279) expression by immunofluorescence staining using confocal microscopy. (**A**) Images of human SH-SY5Y neuroblastoma cells were taken after streptozotocin and menadione (MQ) exposure and post-treatment with vitamin K2. All images were taken with a 63× oil objective. Images were taken in brightfield and FITC modes, where AF 488 (green staining) represents GSK-3 alpha/beta (p-Tyr216, p-Tyr279). Further merged mages were used to observe the differences among control, only VK2, streptozotocin (2.5 mM) and its post-treatment with vitamin K2 (50 µM); menadione (10 µM) and its post-treatment with vitamin K2 (50 µM). (**B**) Representative graphs of the mean fluorescence intensity from three different fields of view using ImageJ software. The data are presented as mean ± SEM. Statistical analysis was performed using Student’s *t*-test and differences with *p*-value ≤ 0.05 were designated as significant. ** *p* < 0.01, and *** *p* < 0.001 denote significant differences compared to the control and between the treatment groups. [STZ—streptozotocin; VK2—vitamin K2; STZ/VK2—post-treatment of streptozotocin (2.5 mM)-treated cells with vitamin K2 (50 µM); MQ—menadione; MQ/VK2—post-treatment of menadione (10 µM)-treated cells with vitamin K2 (50 µM)].

**Figure 8 antioxidants-10-00983-f008:**
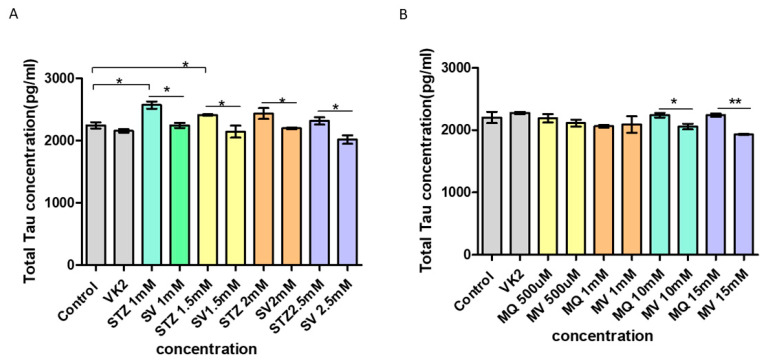
Measurement of total intracellular tau protein levels by ELISA in human SH-SY5Y neuroblastoma cells after (**A**) streptozotocin treatment and post-treatment with vitamin K2 (50 µM) and (**B**) menadione treatment (MQ) and post-treatment with vitamin K2. A best-fit standard curve was plotted for the known protein concentrations using graph pad prism and absorbance values for unknown samples were interpolated on the standard curve to determine the concentration of target protein in each sample. The data are presented as mean ± SEM. Statistical analysis was performed using Student’s *t*-test and differences with *p*-value ≤ 0.05 were designated as significant. * *p* < 0.05 and ** *p* < 0.01 denote significant differences compared to the control and between the treatment groups. [STZ—streptozotocin; VK2—vitamin K2; SV—post-treatment of streptozotocin (1–2.5 mM) treated cells with vitamin K2 (50 µM); MQ—menadione; MV—post-treatment of menadione (500 nM–15 µM)-treated cells with vitamin K2 (50 µM)].

**Figure 9 antioxidants-10-00983-f009:**
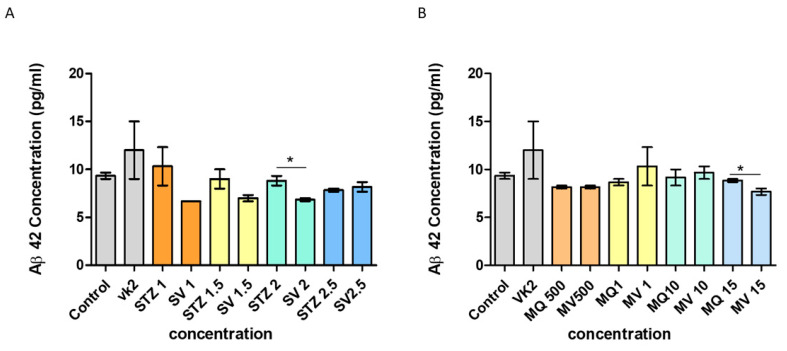
Measurement of secreted Aβ42 protein levels by ELISA in human SH-SY5Y neuroblastoma cells after (**A**) streptozotocin treatment and post-treatment with vitamin K2 (50 µM) and (**B**) menadione treatment (MQ) and post-treatment with vitamin K2. A best-fit standard curve was plotted for the known protein concentrations using Graphpad Prism and absorbance values for unknown samples were interpolated on the standard curve to determine the concentration of target protein in each sample. The data are presented as mean ± SEM. Statistical analysis was performed using Student’s *t*-test and differences with *p*-values ≤ 0.05 were designated as significant. * *p* < 0.05 denote significant differences compared to the control and between the treatment groups. [STZ—streptozotocin; VK2—vitamin K2; SV—post-treatment of streptozotocin (1–2.5 mM)-treated cells with vitamin K2 (50 µM); MQ—menadione; MV—post-treatment of menadione (500 nM–15 µM)-treated cells with vitamin K2 (50 µM)].

**Figure 10 antioxidants-10-00983-f010:**
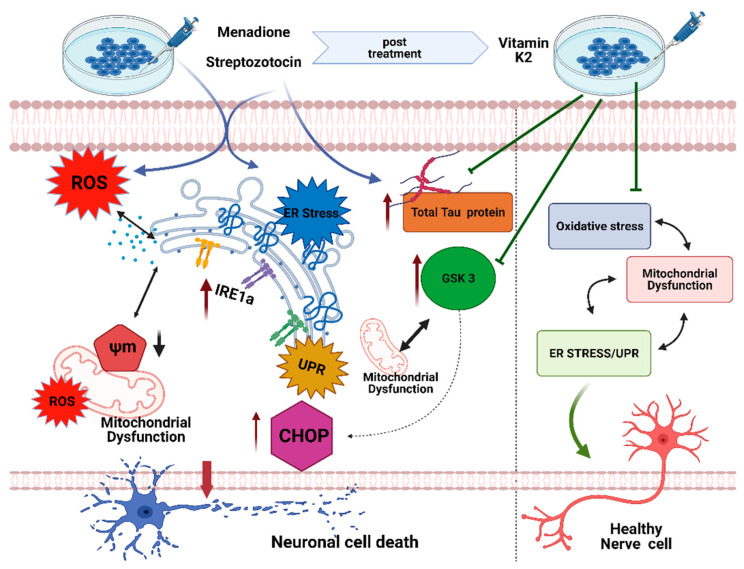
Schematic representation of the protective role of vitamin K2 in streptozotocin- and menadione-induced cytotoxicity in SH-SY5Y cells. Streptozotocin and menadione triggered reactive oxygen species (ROS), which in turn led to mitochondrial depolarization (reduction in mitochondrial membrane potential) and further activated endoplasmic reticulum stress as indicated by the increased expression of CHOP and p-IRE1α. Upregulated CHOP expression is an indication of an increase in proapoptotic signals. Similarly an increase in p-IRE1α level indicated induction of unfolded protein response, which might also result in the induction of proapoptotic signals during prolonged ER stress. Streptozotocin and menadione increased the GSK3 levels, which might be responsible for neuronal cell death by an increase in CHOP expression and mitochondrial dysfunction. In addition to this, streptozotocin treatment also increased the levels of intracellular total tau protein, while menadione was found to have no effect. However, post-treatment of vitamin K2 regulated the neuronal cell death signaling not only by improving mitochondrial health but also by inhibiting ER stress, decreasing the levels of GSK3, and inhibiting the increase in levels of total tau protein. (ROS—reactive oxygen species; ER—endoplasmic reticulum; UPR—unfolded protein response; GSK3—glycogen synthase kinase 3; IRE1—inositol-requiring enzyme; CHOP—C/EBP homologous protein; Ψm—mitochondrial membrane potential.)

## Data Availability

The data will be provided on request.

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
