# Peer review of "Vitamin K2 Modulates Organelle Damage and Tauopathy Induced by Streptozotocin and Menadione in SH-SY5Y Cells"

_antioxidants, 2021, doi:10.3390/antiox10060983_

Round 1

Reviewer 1 Report

Authors did a good job on SH SY5Y characterization in front to streptozocine and menadione, and the role of K2 as antioxidant, mitochondrial function protectant and cytoprotective. Those effects are well known in other in vitro and in vivo models (as de authors claim in the introduction

The most interesting part that is the role of K2 on ER and the downstream pathways up to neurodegenerative hallmarks are less clear and in some cases phenomenological and deserves an improvement in the results presentation.

Methodology

The higher cell survival rate for streptozocine ? or the lower cell survival rate?

Confocal images are not so much convincing (they are blurred and have to be imporved) and a quantification is needed, in different experiments to conclude that GSK3, Ire or Chop change in SH SY5Y cells after insult by streptozocine or menadione, to observe the neuroprotective role of K2

Total ta is not representative of tau phosphorylation, many stimuli can induce a higher tau contact but no a hyperphosphorylation, this point have to be addressed, demonstrating higher tau phosphorylation. To his end a accurate evaluation of GSK3 activation by GSK3b ib Tyr219  will further support the results on tau hyperphosphorylation

In reference to ER, authors claim for the increase in caspase 3 activation or CHOP increase inducing apoptosis that should be diminished by K2. Some experiments addressing the apoptotic cells death in culture can be added (PI nuclei staining is easy and affordable, if you have fixed cultures after immunocytochemistry p.ee))

Otherwise discussions and cartoon have to be changed and avoid overinterpreting results

Author Response

(The corrections are highlighted with yellow shade in the manuscript, as per reviewers’ suggestions)

Reviewer: 1

Comment 1:Authors did a good job on SH SY5Y characterization in front to streptozocine and menadione, and the role of K2 as antioxidant, mitochondrial function protectant and cytoprotective. Those effects are well known in other in-vitroand in-vivomodels (as authors claim in the introduction)

The most interesting part that is the role of K2 on ER and the downstream pathways up to neurodegenerative hallmarks are less clear and, in some cases, phenomenological and deserves an improvement in the results presentation.

Author’s response:Thank you for your nice words recognizing additional finding onthe role of K2 on ER and the downstream pathways up to neurodegenerative hallmarks. To the best of our knowledge this is the first report which shows the role of vitamin K2 in ameliorating ER stress related apoptosis and also its effects on total tau protein and Aβ pathology. All the comments are taken into consideration and improvement have been made accordingly.

Comment 2: Methodology

The higher cell survival rate for streptozocine ? or the lower cell survival rate?

Author’s response:We apologies for such confusion in the line 142-143. Here, this line corresponds to the highest concentration of streptozotocin at which cells are not completely dead, as well as for vitamin K2 it corresponds to the concentration at which it is not toxic. This line might create confusion in readers; therefore, the line has been deleted and rephrased in a simple manner.

line 142-143

“Vitamin K2 was taken in concentrations 1, 10, 15, 25, 50 and 100 µM and for streptozotocin, 1, 1.5, 2- and 2.5-mM concentrations were used”

Comment 3:Confocal images are not so much convincing (they are blurred and have to be imporved) and a quantification is needed, in different experiments to conclude that GSK3, Ire or Chop change in SH SY5Y cells after insult by streptozocine or menadione, to observe the neuroprotective role of K2

Author’s response:Thank you for the comments and suggestions.The quality of the images has been improved and all the images have been set at 300 dpi (as per journal requirements) and few images which were found blurred has also been replaced. In addition, mean fluorescence intensity from three different fields of view and from three independent experiments for each (CHOP, pIRE1 &GSK 3α/β) have been quantified using image J software. Results are presented as mean ± SEM. The quantification of fluorescence intensity clearly depicted the role of vitamin K2 in attenuating the expression of ER stress markers (CHOP & IRE1) and GSK3 α/β. The mean fluorescence intensity graphs are added now in figure 5,6 &7, respectively. Your suggestion for further to confirm the neuroprotective role of vitamin K2 with respect to CHOP, pIRE1 and GSK3 α/β levels, we have planned to conduct it in our future pathway dependent studies, specific to vitamin K2. We will include these studies in the upcoming manuscript.

Comment 4: Total tau is not representative of tau phosphorylation, many stimuli can induce a higher tau contact but no a hyperphosphorylation, this point have to be addressed, demonstrating higher tau phosphorylation. To his end a accurate evaluation of GSK3 activation by GSK3b ib Tyr219 will further support the results on tau hyperphosphorylation

Author’s response: We accept your suggestion.This point was already mentioned in the results section 3.8 line 495-497 “One of the mechanisms underlying Alzheimer's disease is the deposition of the neurofibrillary tangles of the microtubule binding protein tau. About 20 different neurodegenerative diseases are directly linked to “tauopathies”. Therefore, we determined the alterations in level of intracellular total tau protein (irrespective of their phosphorylation state) by ELISA in SHSY5Y cells in response to streptozotocin and menadione treatment followed by the post treatment of vitamin K2”. In the discussion section of the manuscript, (line 678-679) an increased activity of GSK3 is correlated to tau phosphorylation as shown in previous reports, which was cited in support to our study, but our study does not show the direct relation of increased GSK3 activity with the tau phosphorylation, rather it was depicted that it might be possible that increased GSK3 activity is associated with increased total tau protein. Therefore, as suggested and to avoid over representation, this has been corrected in discussion section as well as in conclusion figure (figure 10) of the manuscript. Further, the relation of GSK 3β and tau hyperphosphorylation have been planned to be conducted in our future pathway dependent study specific to vitamin K2.

Comment 5:In reference to ER, authors claim for the increase in caspase 3 activation or CHOP increase inducing apoptosis that should be diminished by K2. Some experiments addressing the apoptotic cells death in culture can be added (PI nuclei staining is easy and affordable, if you have fixed cultures after immunocytochemistry p.ee))

Author’s response: A common method for determining cytotoxicity is based on measuring the activity of cytoplasmic enzymes released by damaged cells or by measuring the disruption of critical biochemical function using MTT assay. Lactate dehydrogenase (LDH) is a stable cytoplasmic enzyme that is found in all cells and its rapid release into the cell culture supernatant in response to plasma membrane damage, is a key feature of cells undergoing apoptosis (Li and Zhang 1997), necrosis (Koh and Choi 1987), and other forms of cellular damage. On the other hand, MTT assay quantifies the mitochondrial activity and critical step in many apoptotic pathways involves the release of Apoptotic inducible factors and cytochrome C from damaged mitochondria, which in turn induces the caspase activation, and eventually to apoptotic cell death. Previous reports also revealed that both LDH release and MTT reduction accurately determine apoptotic death of neurons (Lobner 2000). Thus, we incorporated these two assays in our study, which clearly showed that vitamin K2 was able to reduce the apoptotic cell death induced by menadione and streptozotocin.

This justification has also been included in the manuscript in method section 2.5 line 174-175

“Rapid release of LDH into the cell culture supernatant in response to plasma membrane damage, is a key feature of cells undergoing apoptosis (Lobner 2000)”.

In discussion line 562-565

“MTT assay quantifies the mitochondrial activity and critical step in many apoptotic pathways involves the release of Apoptotic Inducible factors and cytochrome C from damaged mitochondria, which in turn induces the caspase activation, and eventually to apoptotic cell death (Kidd 1998 & Lobner 2000)”

Comment 6:Otherwise discussions and cartoon have to be changed and avoid overinterpreting results

Author’s response: Discussion and the conclusion figure have been changed accordingly and as per suggestion overinterpreted results have been removed.

Reviewer 2 Report

Review of the manuscript entitled “Vitamin K2 modulates organelle damage and tauopathy induced by streptozotocin and menadione in SH-SY5Y cells”. In my opinion the paper is very interesting and well written. However, there are some questions and suggestions that should be considered.

Introduction the introduction is very well written, maybe in line 38 should be added reference.

Methods.

Very important question, were the cells differentiated? I work with SH-SY5Y cells, the correct phenotype of neurons and their features develop after differentiation, e.g. by retinoic acids. If no differentiation has been done, explain why not?

Why was such a very high concentration of H2DCFDA used? Familiarize yourself with article 10.1016/j.pharep.2015.06.005

Very important suggestion. I believe all Figs should be presented as % of control. Control should be 100% and the other variables are higher or lower.

Equally important, some information should be transferred from the results to the discussion section. For example: line 300-303, 327-329, 351-355, 382-389, 396-397, 438-444, 471-473. In the description of the results, we only describe the results, we do not interpret them.

In description of fig 9, explain please and expand the abbreviations as it is in fig 8. Similar in fig 10 legend all abbreviations should be explained.

Author Response

(The corrections are highlighted with yellow shade in the manuscript, as per reviewers’ suggestions)

Reviewer 2

Comment 1: Review of the manuscript entitled “Vitamin K2 modulates organelle damage and tauopathy induced by streptozotocin and menadione in SH-SY5Y cells”. In my opinion the paper is very interesting and well written. However, there are some questions and suggestions that should be considered.

Author’s response: Thank you for your kind words and recognizing frontier research area and potential future use. All your comments and suggestions have been considered throughout the manuscript.

Comment 2: Introduction the introduction is very well written, maybe in line 38 should be added reference.

Author’s response: Thanks for nice words and as suggested, the reference has been added in line 38 (Wimo et al., 2003).

Comment 3:Methods:- Very important question, were the cells differentiated? I work with SH-SY5Y cells, the correct phenotype of neurons and their features develop after differentiation, e.g. by retinoic acids. If no differentiation has been done, explain why not?

Author’s response: Great question, appreciated!As mentioned in the introduction section, line 113-114 “SH-SY5Y cells were used in this study because of their close resemblance to neurons and is widely accepted model for neurodegenerative diseases [30,31]”.Recent reports have also shown the use of the human neuroblastoma (SH-SY5Y) cells as a dopaminergic neuronal model for neuronal disorders such as Parkinson’s disease (PD) (Jaafaru et al., 2018) , Alzheimer’s disease (AD) (Oboh et al., 2013), Huntington’s disease (HD) (Ismail et al., 2014), multiple sclerosis (MS) and Amyotrophic lateral sclerosis (ALS) (Kovalevich and Langford, 2013). Therefore, SHSY5Y cells as a neuronal model for many neuronal disorders were used to study the neuroprotective effect of vitamin K2 in response to streptozotocin and menadione. Also, SHSY5Y cells (undifferentiated) were used in this study to understand the role of vitamin K2 in modulating the basal levels of total tau protein and Aβ42. Our intension was to start with SH-SY5Y cells first to setup a base for future studies with differentiated cell line specific to AD. To the best of our knowledge, this is the first study together with vitamin K2 against streptozotocin and menadione response, which we will continue in our future study, where we have a plan to evaluate the differences in the neuroprotective effect of vitamin K2 in undifferentiated and differentiated SHSY5Y cells, because many recent reports have indicated the differences in response to treatment in case of undifferentiated and differentiated SHSY5Y cells (Bagaméry et al., 2021).

Comment 4:Why was such a very high concentration of H2DCFDA used? Familiarize yourself with article 10.1016/j.pharep.2015.06.005

Author’s response: Thanks for familiarizing with this useful article. Reactive oxygen species measurement done using H2DCFDA in this study was performed as a standardized protocol used previously in our group (Kesari et al. 2015: https://doi.org/10.1016/j.mrgentox.2015.10.004)and several others (Luukkonen et al., 2014: https://doi.org/10.1016/j.mrfmmm.2013.12.002; Luukkonen et al. 2009: https://doi.org/10.1016/j.mrfmmm.2008.12.005). The concentration of DCFDA used in this study was same as utilized in our previous study (Kesari et al., 2020).

In context with the article 10.1016/j.pharep.2015.06.005 as suggested, though Tetrabromobisphenol A (TBBPA) was found to interact with H2DCFDA and enhanced the fluorescence signal in a cell free model, many reports have indicated the streptozotocin and menadione induces ROS in vitrobut were not found to interact with H2DCFDA in cell free model.

Comment 5:Very important suggestion. I believe all Figs should be presented as % of control. Control should be 100% and the other variables are higher or lower.

Author’s response: Thanks for this valuable suggestion, this can be valid only for the cell viability assay where the control can be 100% viable cells and other variables are considered higher or lower with respect to it. However, the percentage viability graphs depict the same result as RAU (Relative Absorbance Unit), therefore RAU was considered. Whereas in other assays, like ROS, LDH and JC1 assay control could not be considered as 100%, because control involves untreated cells. Similar data presentation has been reported from our group along with several others.

Comment 6:Equally important, some information should be transferred from the results to the discussion section. For example: line 300-303, 327-329, 351-355, 382-389, 396-397, 438-444, 471-473. In the description of the results, we only describe the results, we do not interpret them.

Author’s response: We agree with your comment andas per the suggestion the information from the results have been transferred to the discussion and highlighted.

Comment 7:In description of fig 9, explain please and expand the abbreviations as it is in fig 8. Similar in fig 10 legend all abbreviations should be explained.

Author’s response: As per the suggestion, the figure 9 legend has been explained properly and abbreviations are expanded (line 528-530). Similarly, legend has been explained properly and abbreviations are expanded in figure 10 (line 723-735).

References:

Lobner, D., 2000. Comparison of the LDH and MTT assays for quantifying cell death: validity for neuronal apoptosis?. J. Neurosci. Methods96(2), pp.147-152.

Koh JY, Choi DW., 1987. Quantitative determination of glutamate mediated cortical neuronal injury in cell culture by lactate dehydrogenase efflux assay. J Neurosci Meth, 20, pp. 83–90.

Li J, Zhang J., 1997, Inhibition of apoptosis by ginsenoside Rg1 in cultured cortical neurons. Chin Med J (Engl),110, 535–9.

Kidd VJ.,1998, Proteolytic activities that mediate apoptosis. Annu Rev Physiol, 60, 533–73.

Jaafaru, MS., Nordin, N., Shaari, K., Rosli, R., Razis, A.F.A. 2018, Isothiocyanate from Moringa oleifera seeds mitigates hydrogen peroxide-induced cytotoxicity and preserved morphological features of human neuronal cells. PLoS ONE, 13, e0196403.

Oboh, G., Agunloye, OM., Akinyemi, AJ., Ademiluyi, AO., Adefegha, SA. 2013, Comparative study on the inhibitory effect of caffeic and chlorogenic acids on key enzymes linked to Alzheimer’s disease and some pro-oxidant induced oxidative stress in rats’ brain-in vitro. Neurochem. Res. 38, 413–419.

Ismail, N., Ismail, M., Imam, MU., Azmi, NH., Fathy, SF., Foo, JB., Bakar, MFA. 2014, Mechanistic basis for protection of differentiated SH-SY5Y cells by oryzanol-rich fraction against hydrogen peroxide-induced neurotoxicity. BMC Complem. Altern. Med. 14, 467.

Kovalevich, J., Langford, D. 2013, Considerations for the Use of SH-SY5Y Neuroblastoma Cells in Neurobiology. Neuronal Cell Cult; Humana Press: Totowa, NJ, USA, pp. 9–21.

Bagaméry, F., Varga, K., Kecsmár, K., Vincze, I., SzökÅ‘, É. and Tábi, T., 2021. The Impact of Differentiation on Cytotoxicity and Insulin Sensitivity in Streptozotocin Treated SH-SY5Y Cells. Neurochem Res46(6), pp.1350-1358.

Delacourte, A., 2005. Tauopathies: recent insights into old diseases. Folia Neuropathologica43(4).

Wimo, A., Winblad, B., Aguero-Torres, H. and von Strauss, E., 2003. The magnitude of dementia occurrence in the world. Alzheimer Disease & Associated Disorders17(2), pp.63-67.

Kesari, KK., Dhasmana, A., Shandilya, S., Prabhakar, N., Shaukat, A., Dou, J. Rosenholm, JM. Vuorinen, T., Ruokolainen, J. 2020, Plant-derived natural biomolecule picein attenuates menadione induced oxidative stress on neuroblastoma cell mitochondria. Antioxidants.,9, 1–17, doi:10.3390/antiox9060552.

Round 2

Reviewer 1 Report

I disagree with authors because, as they claim LDH measure membrane rupture, and this can occurs both after necrotic process and when apoptotic cells are definitely death. It is not indicative actually for apoptotic process,

They can be a process of apoptosis before the membrane rupture. Then in my opinion LDH is not a suitable marker of a cell undergoing into apoptosis but of definitively cell death.

I suggest to moderate the affirmation that they are measuring apoptosis.

In line 610, again MTT and mitochondrial function measured alteration in mitochondria and the dysfunction can be the initial point of apoptotic signaling , but in fact none of both methods measure if this process is initiated or active, albeit Lobster propose those as a apoptotic measures. The sentence, and other in discussion have to be rewrote in this sense, because no direct parameters of apoptosis were measured.

The other issues has been answered adequately

Author Response

Author’s response to reviewer comments

Reviewer 1:

I disagree with authors because, as they claim LDH measure membrane rupture, and this can occurs both after necrotic process and when apoptotic cells are definitely death. It is not indicative actually for apoptotic process,

They can be a process of apoptosis before the membrane rupture. Then in my opinion LDH is not a suitable marker of a cell undergoing into apoptosis but of definitively cell death.

I suggest to moderate the affirmation that they are measuring apoptosis.

In line 610, again MTT and mitochondrial function measured alteration in mitochondria and the dysfunction can be the initial point of apoptotic signaling , but in fact none of both methods measure if this process is initiated or active, albeit Lobster propose those as a apoptotic measures. The sentence, and other in discussion have to be rewrote in this sense, because no direct parameters of apoptosis were measured.

The other issues has been answered adequately

Author’s response: Thanks for this valuable suggestion. As per your suggestions, we have corrected in the text. We agree with you that none of both methods measures the direct apoptotic process, is initiated or active. Therefore, in consideration to your suggestion and to avoid any confusion in the manuscript, the affirmation of measuring apoptosis have been corrected, because both MTT and LDH are the appropriate measures of quantifying cell death. Changes have been made in the discussion and overall conclusion of the study as well as in conclusion figure. We have highlighted all the changes in the text with blue color shade.